# Has the COVID-19 Pandemic Led to Changes in the Tasks of the Primary Care Workforce? An International Survey among General Practices in 38 Countries (PRICOV-19)

**DOI:** 10.3390/ijerph192215329

**Published:** 2022-11-20

**Authors:** Peter Groenewegen, Esther Van Poel, Peter Spreeuwenberg, Ronald Batenburg, Christian Mallen, Liubove Murauskiene, Antoni Peris, Benoit Pétré, Emmily Schaubroeck, Stefanie Stark, Emil L. Sigurdsson, Athina Tatsioni, Kyriaki Vafeidou, Sara Willems

**Affiliations:** 1Netherlands Institute for Health Services Research (NIVEL), 3500 BN Utrecht, The Netherlands; 2Department of Sociology, Utrecht University, 3584 CS Utrecht, The Netherlands; 3Department of Human Geography, Utrecht University, 3584 CS Utrecht, The Netherlands; 4Department of Public Health and Primary Care, Ghent University, 9000 Ghent, Belgium; 5Department of Sociology, Radboud University, 6535 XN Nijmegen, The Netherlands; 6School of Medicine, Keele University, Keele ST5 5BG, UK; 7Department of Public Health, Faculty of Medicine, University of Vilnius, 03101 Vilnius, Lithuania; 8Castelldefels Agents de Salut (Casap), 08860 Castelldefels, Spain; 9Department of Public Health, Faculty of Medicine, University of Liège, 4000 Liège, Belgium; 10Institute of General Practice, Friedrich-Alexander University Erlangen-Nürnberg (FAU), 91054 Erlan-gen-Nürnberg, Germany; 11Department of family medicine, University of Iceland, 102 Reykjavík, Iceland; 12Research Unit for General Medicine and Primary Health Care, Faculty of Medicine, 45110 Ioannina, Greece

**Keywords:** primary health care, general practice, general practitioners, task changes, quality of care, primary care workforce, COVID-19, international comparison, PRICOV-19

## Abstract

The COVID-19 pandemic has had a large and varying impact on primary care. This paper studies changes in the tasks of general practitioners (GPs) and associated staff during the COVID-19 pandemic. Data from the PRICOV-19 study of 5093 GPs in 38 countries were used. We constructed a scale for task changes and performed multilevel analyses. The scale was reliable at both GP and country level. Clustering of task changes at country level was considerable (25%). During the pandemic, staff members were more involved in giving information and recommendations to patients contacting the practice by phone, and they were more involved in triage. GPs took on additional responsibilities and were more involved in reaching out to patients. Problems due to staff absence, when dealt with internally, were related to more task changes. Task changes were larger in practices employing a wider range of professional groups. Whilst GPs were happy with the task changes in practices with more changes, they also felt the need for further training. A higher-than-average proportion of elderly people and people with a chronic condition in the practice were related to task changes. The number of infections in a country during the first wave of the pandemic was related to task changes. Other characteristics at country level were not associated with task changes. Future research on the sustainability of task changes after the pandemic is needed.

## 1. Introduction

The COVID-19 pandemic has had a large impact on primary care (PC) [1]. This impact varies by country, providing the opportunity to learn from each other [2,3]. General practices (GP practices) across Europe have adapted their organization [4,5]. These adaptations are multiple and concern ways of dealing with specific challenges in the care for COVID-19 patients, controlling patient flows, keeping regular care going, implementing procedures for infection prevention, etc. As part of these organizational adaptations, general practitioners (GPs) and practice staff have had to take up new tasks or take tasks over from other professionals, such as hospital staff. This paper focuses on the changes in the tasks of GPs and staff during COVID-19. We use data from the international PRICOV-19 study of GPs in 37 European countries and Israel. This international collaborative study, initiated by Ghent University (Belgium), aims to assess the impact of the COVID-19 pandemic on the organization of care and the different dimensions of quality of care in GP practices [6]. 

Task changes in general practice relate to the broader concept of task shifting. Task shifting is well established and research in this area provides useful background to the study of task changes in general. Task shifting is defined as the reallocation and redistribution of tasks and the sharing of roles among health professions and different groups of health professionals [7,8]. Shifting tasks from GPs to other professionals, such as nurses and support staff, is usually a slow and gradual process. However, with the COVID-19 pandemic, changes in who performed which tasks rapidly changed. This raises new questions about the circumstances under which externally forced changes in tasks were possible and, ultimately, about the sustainability of the changes.

During the pandemic, the tasks of GPs and staff have been intensified and/or shifted. Practice staff, such as practice assistants/secretaries and nurses, have been allocated new tasks. The type of staff employed by a practice differs between countries and practices [9,10]. Whilst volunteers also play a role in GP practices, our analysis only includes paid staff. We will use the term ‘changes in tasks’ instead of the more common and related term ‘shifts of tasks’ because the changes also include intensifying existing tasks and taking up new responsibilities. 

The added value of our analysis is the international comparison. Therefore, we will distinguish between barriers and facilitators at the level of the health systems and countries on the one hand and the level of the practices on the other. Moreover, we distinguish between the specific situation of the pandemic and the general influences known to impact task shifting in PHC.

### 1.1. Country Level

As *specific* influences on changes in tasks at the level of the countries/health systems, we assume that the urgency of changes may have played a role; this is related to the intensity of the COVID-19 pandemic in each country in terms of, e.g., numbers of infections and mortality during the first wave of the pandemic. Also, the role of PC during the pandemic may have had an impact on the urgency of task changes (see, e.g., [10] on the role of GPs in Social Health Insurance systems). When GP practices were the first point of contact for people with symptoms of COVID-19 infection, or had a role in testing, the workload may have increased quickly, necessitating changes in tasks. An example of this is issuing ‘sick notes’. In Germany, e.g., procedures changed during the pandemic; sick notes can be requested by telephone, assistants prepare everything, and the GP only signs. In Lithuania, the requirements for writing out sick leaves were simplified from March 2020 to May 2022, and the National Public Health Center was tasked with administering sick leave for specific groups of people. In the United Kingdom, the duration of the period people could ‘self-certify’ increased, removing work from GPs. Self-certification was introduced in Catalonia (Spain) from February 2022. In Spain, telephone calls were formerly handled by an external call center. However, this was stopped during the pandemic and moved to clerks at the GP practices who had to be prepared for this new task. Payment systems for GP care may affect the willingness to shift tasks in practices with a fee-for-service payment model if, e.g., there are not yet tariffs for consultations with nurses.

There are also *general* influences on changes in tasks that are independent of the specific situation of the pandemic. First of all, there are significant differences between European health systems in the extent to which tasks have already been shifted to nurses and support staff in PC. For example, a secondary analysis of the QUALICOPC study, with data collected around 2012, showed that England and Sweden scored high on task shifting to nurses and support staff, while Austria and Switzerland scored low [11]. This starting point could lead to few changes in tasks in some countries and major ones in others; it is conceivable that where tasks have already been shifted to nurses and support staff in the past, the pandemic does not change much; the workload of nurses and support staff may have increased, but maybe no new tasks have been taken up by them. However, where task shifting is not so well-developed, the need for allocating new or changed tasks to nurses and support staff may be much higher. 

At the country level, the way the health system and, in particular, PC is organized and resourced may also play a role in the changes in tasks during the pandemic [12]. In countries with a stronger PC system (characterized among other things by a gate-keeping function and a defined patient list), task shifting may be further developed, and changes in tasks and responsibilities may be easier to implement compared to countries with weaker PC. Institutions such as the education of nurses and support staff in PC, legal barriers to task shifting, professionalization of nurses, and policy and ideology regarding task shifting [11], make task changes more normal and also may affect the extent to which tasks have changed during the pandemic. 

### 1.2. Practice Level 

At the level of the practices, there may be *specific* reasons to implement changes in tasks. One reason may be the problems due to staff absence because of COVID-19 infection, shielding, or quarantine requirements. In *general*, larger practices increase the possibility of new job arrangements; in single-handed practices, there are fewer people to shift tasks to, while there are more opportunities in multidisciplinary practices. Adaptations to practice organization in Spain underscore the potential of existing multidisciplinary teams for task-shifting, expanded roles and new team members [13]. Changes in tasks may be affected by the environment in which GP practices work. Urban or rural location and the composition of the practice population, e.g., in terms of social deprivation or minority groups, may be related to changes in tasks.

### 1.3. Aim and Research Questions

Against this background, we aim to gain insight into changes in tasks in the GP practices during the COVID-19 pandemic of the 38 participating countries in the PRICOV-19 study. Our first research question is descriptive: to what extent have the tasks of non-GP staff (practice assistants/nurses) changed during the pandemic, and to what extent have GPs taken up more responsibilities during the pandemic, and how do they evaluate this? Our second question relates to the variation between practices and countries in these respects: how can we explain variation in changes in tasks of non-GP staff and responsibilities of GPs?

## 2. Methods

### 2.1. Study Design and Setting

In the summer of 2020, an international consortium of more than 45 research institutes was formed under the coordination of Ghent University (Belgium) to investigate how GP practices were organized during the COVID-19 pandemic (PRICOV-19). PRICOV-19 uses a cross-sectional study design. Data are collected in 37 European countries (Austria, Belgium, Bosnia and Herzegovina, Bulgaria, Croatia, Cyprus, Czech Republic, Denmark, Estonia, Finland, France, Germany, Greece, Hungary, Iceland, Ireland, Italy, Kosovo*, Latvia, Lithuania, Luxembourg, Malta, Moldova, North Macedonia, Norway, Poland, Portugal, Romania, Serbia, Slovenia, Spain, Sweden, Switzerland, The Netherlands, Turkey, United Kingdom and Ukraine) and in Israel. 

Data were collected by means of an online self-reported survey among GP practices. The survey was developed at Ghent University in multiple phases, including a pilot study among 159 GP practices in Flanders (Belgium). More details are described in the protocol [6]. The survey consisted of 53 items arranged across six topics: (a) infection prevention; (b) patient flow for COVID- and non-COVID care; (c) dealing with new knowledge and protocols; (d) communication with patients; (e) collaboration and wellbeing of the respondent; (f) and characteristics of the respondent and the practice. 

The survey was translated into 38 languages following a standard procedure. The Research Electronic Data Capture (REDCap) platform was used to host the survey in all languages, send out invitations to the national samples of general/family practices and securely store the answers from the participants [14]. Survey data are complemented with data on the country’s healthcare system, the regional and national measurements taken during the COVID-19 pandemic, and the impact of COVID-19 on the country’s population health from the Organization for Economic Cooperation and Development (OECD) (see: The impact of COVID-19 on health and health systems—OECD) and the ‘COVID-19 Health System Response Monitor’ (HSRM) (see: Overview (who.int)).

### 2.2. Sampling and Recruitment

Data were collected between November 2020 and October 2021. Duration of data collection varied between countries (3 to 35 weeks). In each partner country, the consortium partner(s) recruited general/family practices following a predefined recruitment procedure [6]. One survey was completed per practice, preferably by a GP or staff member familiar with the practice organization (e.g., GP practice owner). At least one reminder was sent in all countries. 

The research team at Ghent University collected, stored and cleaned the survey data. Partners received their national database for country-specific analyses and could participate in the international analysis. The protocol of this study and the data handling protocols are described in the Data Management Plan registered at Ghent University. 

The country partners provided additional information on the recruitment strategies and sampling procedures. Moreover, they gave (estimates) of the population distribution according to urbanization of the practice location (big city, suburbs and small towns, semi-rural and rural), the size of the practice (number of patients in five categories) and the practice type (solo, duo, group) in order to be able to assess representativeness of the participating GPs. 

### 2.3. Measurements

The English language version of the questionnaire is included as an additional file in the protocol paper [6]. The following variables were used in this paper.

Dependent variables: 

Changes in the tasks of non-GP staff have been measured using four items:(1)Staff members are more involved in giving information and recommendations to patients contacting the practice by telephone.(2)Staff members are more involved in giving information or explaining what a caregiver has said to illiterate patients, patients with low health literacy or migrants.(3)Staff members are more involved in actively reaching out to patients that might postpone healthcare.(4)Staff members are more involved in the triage.

Changes in the tasks of the respondent (usually a lead GP or GP practice owner) have been measured using two items: (1)My responsibilities in this practice increased.(2)Since the COVID-19 pandemic, GPs or GP trainees are more involved in actively reaching out to patients that might postpone healthcare.

Practice-level independent variables:Staff absence using the question: Since the COVID-19 pandemic, how many staff members had to take time off in practice due to COVID-19 (because of being infected or because of being in quarantine)?Coping with absenteeism of practice staff measured by three items about coping internally, coping in cooperation with neighboring practices, and improved cooperation with neighboring practices. These ways of coping can be a way to mitigate the pressures of the COVID-19 pandemic in general or of absenteeism in practice.GPs’ evaluation of their role changes through three items: I am happy with the task shifting in my professional role; I don’t feel prepared for the task shifting in my professional role; I need further training for these amended responsibilities.Practice size by the question: How many patients are registered in this practice? If there is no registration, please indicate the total practice population. Outliers were recoded to the tail of the distribution.Number of GPs and trainees by the question: How many GPs and GP trainees are working in this practice? Outliers were recoded to the tail of the distribution.Number of disciplines working in the practice using the total number of different disciplines, based on the question about the different disciplines working in the practice.Total number of paid staff by the question: How many people work in this practice? Outliers were recoded to the tail of the distribution.The payment system of GPs was entered as a dummy variable for fee-for-service and mix of fee-for-service and other payment elements vs. other payment systems.The practice location of the practice measured by the following options: big (inner) city, suburbs, (small) town, mixed urban–rural, rural.The composition of the practice measured along the following dimensions: Patients with a migration background, patients with limited health literacy or low literacy, patients who live in poverty, patients with a psychiatric vulnerability, patients over the age of 70, patients with chronic conditions, patients with little social support or limited informal care. The answering options were: below average, approximately the average, and above average, I do not know. Based on the correlations, we have combined the dimensions of patient age and chronic conditions into one variable and the other dimensions in another variable (sum).

Country level independent variables:The numbers of infections and mortality during the first wave of the pandemic and (as an alternative) during the three months before the start of the data collection (source: ECDC for EU countries; national coordinators for countries outside EU).The role of GP practices during the pandemic, asked in a separate survey to all partners in the PRICOV-19 study. More specifically, this focused on the following areas: testing, manning the testing sites, contact tracing, writing sickness absence certificates, writing quarantine certificates, care for/treatment of COVID-19 patients, the vaccination campaign (with answering options yes, no, I don’t know and not applicable). In addition, we have created a new variable by summing the tasks of GPs.The extent to which tasks have already been shifted to staff in PC: scores on task shifting to staff by country from the QUALICOPC study [11].Strength of PC: data from the PHAMEU project [15]. This variable was built from indicators in three dimensions: governance of primary care, workforce development and economic conditions for primary care, with values ranging from 1 (weak primary care) to 3 (strong primary care).Institutional factors: whether nurses have prescription rights in a country. Using data from Kroezen et al. and Maier [16,17], we classified countries into two categories: 1 = no prescription rights (Austria, Belgium, Bulgaria, Czech Republic, Germany, Greece, Hungary, Iceland, Italy, Latvia, Lithuania, Luxemburg, Malta, North Macedonia, Portugal, Rumania, Slovenia, Turkey); 2 = prescription rights (Cyprus, Denmark, Estonia, Finland, Ireland, Netherlands, Norway, Poland, Spain, Sweden, one canton in Switzerland, United Kingdom). The other eight countries were missing on this variable. This variable was previously used in Groenewegen et al. [11].

### 2.4. Statistical Analysis

To examine how well the distribution of the total population of GP practices according to practice size, location and practice type for each country, as reported by country coordinators, reflected the distribution of these characteristics in the participating GP practices in the PRICOV-19 study, we used the standard approach to conduct a one-sample chi-square test [18].

We described the main variables used in the analysis in tables by country without any statistical tests. For ease of reading, we divided the average responses per country into three equal groups and used dark grey shading for the highest tertile, light grey for the middle tertile and no shading for the lowest tertile. 

Item non-response occurred because some respondents did not complete the online survey or skipped questions. This problem was tackled by including dummy variables for (partial) missing data. In doing so, we tested if a non-significant relationship between the missing-value indicator and dependent variable implies that the missing values were not related to the outcome—and hence it was most probably missing at random. As a sensitivity analysis, all analyses were re-run using the dataset with complete cases only. We have also run two additional sensitivity analyses. The first was to take into account the timing of filling out the questionnaire, which differs between countries and respondents, and which may have influenced changes in tasks (added variable at GP level: time between the start of the pandemic (February 2020) and the month that the questionnaire was filled out). The second sensitivity analysis was to take into account the position of the respondent (GP or trainee GP compared to other positions). 

The analysis was carried out using multilevel analysis to account for the nested structure of the data [19]. Inspection of the correlations between the independent variables at GP level showed only two correlations above 0.70 (total number of staff with practice size0.76and with absenteeism 0.72). These correlations are not so high as to influence the regression coefficients. 

A task-changes scale was developed in a three-level ecometric or latent variable model instead of aggregating a scale from the GP level to the country level. The advantages of the ecometric approach are that the two-level structure of the data is taken into account, missing answers to some of the items can easily be handled, and that scale properties, such as reliability, can be defined at the level of GPs but also at the level of countries. In the ecometric analysis, the scale items are at the lowest level in a multi-response model. The six items are combined into a scale representing the latent variable ‘task changes’. The resulting scale can be considered as a continuous variable. Scale reliability, which is equivalent to Cronbach’s alpha in single-level models, is calculated as: the country-level variance/country-level variance + (practice-level variance/average number of respondent per country) + the sum of the item level variances/(number of items times the average number of respondents per country). The item-level variance is also known as the measurement error. Scale values are scored such that the scale average reflects the original five-point answering categories. For background to ecometric analysis, we refer to [20].

We used random effects (variances) at the country level to describe the clustering of task changes during the pandemic. Due to the small number of countries involved, we included country-level variables one at a time for the statistical analysis. Boundary values of *p* < 0.05 were set for statistical significance at the GP level and *p* < 0.10 for the country level.

The modelling strategy consisted of the following steps:Empty model to calculate the clustering of the dependent variables within countries;Adding GP/practice variables;Adding interaction terms for the interaction between staff absence and coping mechanisms to model the possibility that staff absence is less problematic when adequately coped with;Adding country variables (one by one).

Analyses were performed in MLwiN (version 2.30, Centre for Multilevel Modelling, University of Bristol, UK).

### 2.5. Ethical Approval

The study was conducted according to the guidelines of the Declaration of Helsinki. The Research Ethics Committee of Ghent University Hospital approved the protocol of the PRICOV-19 study (BC-07617). Research ethics committees in the different partner countries gave additional approval if needed in that country, as included in the protocol paper [6]. All participants gave informed consent on the first page of the online survey.

## 3. Results

The recruitment strategies used by the country partners included an invitation published on a website or social media (2 countries), direct contact to all GP practices (12 countries), contact with a sample of GP practices (10 countries), other strategies (3 countries), and multiple recruitment strategies (11 countries). Random sampling was the preferred approach of the study protocol, but in only 6 countries was this realized; in 21 countries a convenience sample was used, and in 11 countries a mix of sampling approaches was used.

Participation rates and representativeness are discussed in the protocol paper (in Table 1 respectively) [6], and in [21]. The participation rate was 22% (IQR 10-28), varying from below 10% in, amongst others, Denmark, Sweden and Latvia, to 90% or higher in Serbia, North Macedonia, Greece and Bulgaria. Overall, the participating practices did not mirror the population of practices in the countries in terms of the distribution by urbanization of the practice location, the size of the practice and practice type. Practices in small towns and suburbs were under-represented and those in (semi-)rural areas were over-represented, as were large practices (over 10,000 patients) and group practices. 

### 3.1. Scale Analysis

The scale analysis showed that the reliability at the country level was high (0.95) and at the GP level satisfactory (0.77). Its clustering was considerable—ICC = 20.4, meaning that 20% of the variation in task changes is at the country level and 80% at the level of the GP practices. The distribution of the scale by country is provided in Appendix A.

### 3.2. Task Changes

The first four variables in the task-change scale related to tasks that have been shifted to or intensified by the practice staff. The percentage agreeing and strongly agreeing with the four items are given in Table 1 per country. Overall, the highest percentages of agreeing and strongly agreeing were for staff members being more involved in giving information and recommendations to patients contacting the practice by phone, and for staff being more involved in triage patients. If we only look at the percentage strongly agreeing, four countries stand out with the highest percentage for all four items: Italy (69% for giving information by telephone; 70% for explaining to patients with low health literacy; 46% for reaching out; and 56% for involvement in triage), Latvia (with 71%, 60%, 67% and 75%, respectively), Lithuania (with 54%, 45%, 23% and 56%, respectively) and Romania (with 51%, 48%, 47% and 55%, respectively). On the other side, the lowest percentages strongly agreeing were reported from GP practices in Bosnia and Herzegovina (13% for giving information by telephone; 13% for explaining to patients with low health literacy; 10% for reaching out; and 10% for involvement in triage), Finland (with 5%, 31%, 10% and 5%, respectively), Greece (with 20%, 19%, 13% and 24%, respectively), Israel (with 12%, 5%, 10% and 20%, respectively) and Serbia (with 12%, 11%, 12% and 20%, respectively). 

Two variables were used to construct the scale changes in the tasks of the respondent (usually a lead GP or GP practice owner) (see Table 2). Our respondents most often strongly agreed that their responsibilities had increased in Lithuania (strongly agree 78%), Latvia (strongly agree 76%) and Romania (strongly agree 74%); with the lowest percentages in North Macedonia (strongly agree 0%), Iceland (strongly agree 19%) and Israel (strongly agree 14%). Another aspect of changing tasks for GPs and trainees is that they might be more involved in actively reaching out to patients who might postpone healthcare. Respondents in Italy (strongly agree 68%), Latvia (strongly agree 64%) and Romania (strongly agree 49%) most often strongly agreed that this was the case, while among respondents from Finland (strongly agree 2%), Malta (strongly agree 0%) and Luxemburg (strongly agree 0%), this was least often the case.

The evaluation and satisfaction with the increase in responsibilities are shown in Table 3. Respondents in Kosovo* (strongly agree 40%), Iceland (strongly agree 23%) and Moldova (strongly agree 16%) most often strongly agreed with the statement that they were happy with the task shifting in their professional role. Respondents from Kosovo* (strongly agree 19%), Ukraine (strongly agree 12%) and North Macedonia (strongly agree 10%) most often strongly agreed that they did not feel prepared for the task shifting in their professional role. Finally, respondents in Kosovo* (strongly agree 19%), Romania (strongly agree 18%) and Turkey (strongly agree 16%) most often strongly agreed that they needed further training for these amended responsibilities.

### 3.3. (Coping with) Absenteeism of Staff

Changes in tasks may have been influenced by the pressure of absenteeism of practice staff due to a COVID-19 infection or quarantine. Absenteeism was difficult to compare in absolute numbers because of differences in the size of the practices. However, weighted for the number of paid staff, absenteeism had been relatively high in GP practices in Serbia, Kosovo* and Croatia, and relatively low in Italy, Estonia and Finland (Table 4). We also asked how practice teams have coped with this. Coping with absenteeism of practice staff (if staff members in this practice are absent because of COVID-19 (infection or quarantine), the work can be distributed in such a way that the well-being of colleagues is not compromised; this practice can count on the help of other PC practices in the neighborhood, or improved cooperation with neighboring practices) can be a way to mitigate the pressures of the COVID-19 pandemic in general or of absenteeism in practice. The option of internal coping was most often strongly agreed with in Latvia (53%), Kosovo* (50%) and Moldova (30%); coping with neighboring practices was most often agreed with among GPs in Kosovo* again (47%), Latvia (32%) and Slovenia (also 32%). Finally, in again Kosovo* (46%), the UK (29%) and Moldova (23%), GPs strongly agreed that the cooperation with neighboring practices had improved.

Information on the size of the practices in terms of numbers of patients, GPs and staff is in Appendix A.

### 3.4. Country and Health System Characteristics

Information on the country and health system characteristics used in the analysis is provided in Appendix A.

### 3.5. Results of the Statistical Analysis

Table 5 includes the results of the multilevel regression analyses. The random part of the models demonstrates that a quarter of the variation in task changes of GPs and practice staff was located at the country level, and three quarters represented differences between practices (ICC = 25.7% in Model 1). Model 2 shows that task changes were related to improved cooperation with neighboring practices. In practices with more task changes, GPs were happy with task changes but also indicated a need for further training. The number of different disciplines working in a practice was related to more task changes. An above-average proportion of the elderly and people with chronic conditions in the practice population also coincided with more task changes. 

Model 3 (Table 5) shows a significant interaction between staff absence and coping internally with staff shortage. Practices with more absence of their staff due to COVID-19 or quarantine and who cope with this internally had changed more tasks. Overall, comparing the empty model (Model 1) with the model including all practice-level variables (Model 3), there was only a small variance decrease at the practice-level (3.5%).

The variables at the country level were introduced one by one due to the (relatively) small number of countries. Model 4 (Table 5) shows that none of the country characteristics reached statistical significance; however, the number of COVID-19 cases per million inhabitants during the first wave came close to significance at *p* = 0.10, the level set for the country-level variables. However, this variable has no explanatory power (based on the country-level variation with the number of COVID-19 cases per million inhabitants during the first wave included (not in Table 5), compared with the country variance in the empty model (Model 1).

As we pointed out in the methods section, we have added missing-values indicators to keep as many observations as possible. Some of these missing-values indicators were significantly related to task changes. This indicates that missing data were not completely random, which is why we conducted a sensitivity analysis based on complete GP data only (Appendix A). The results were in general the same; the coefficients of the variables on the evaluation of task changes by GPs (GPs happy with task changes and need for further training) were not significant in the sensitivity analysis with only complete data. However, the coefficient of the number of COVID-19 cases per million inhabitants during the first wave was significant in the sensitivity analysis. 

We have carried out two additional sensitivity analyses. The first one was to account for the time between the start of the pandemic and filling out the questionnaire. This variable showed no relation to the outcome and did not affect the relationship between the other independent variables and the outcome (not in table). The second addressed the position of the respondent within the practice. This varied between countries. Overall, 93% were GPs (or trainee GPs), with this being as high as 100% in Denmark, Iceland, Ireland and Romania, and as low as 28% in Sweden, 70% in Italy, 75% in Greece and 78% in Moldova. In a sensitivity analysis with an additional variable to indicate the position of the respondent (GP or trainee GP compared to other positions), no relation to the outcome was found and the relationship between the other independent variables and the outcome was not affected (not in table). 

## 4. Discussion

### 4.1. Main Findings 

Task changes for practice staff and GPs were the core dependent variable in our analysis. This was addressed in our first, descriptive research question. Staff members were more involved in giving information and recommendations to patients contacting the practice by phone. In addition, they were more involved in the triage of patients. GPs in all countries experienced more responsibilities and involvement in reaching out to patients. These findings show the impact of the pandemic on the day-to-day work of GPs and their staff. Behind the overall picture of changes are large differences between countries.

Our second research question was how we could statistically explain these differences in task changes. 

In the introduction, we identified potential general and specific influences on task changes at practice and country level to guide the choice of variables in the statistical analysis. At the practice level, these referred to problems due to absence of staff, the size of the practices, and the practice environment and patient population. The statistical analysis found that problems due to staff absence because of COVID-19 infection, shielding, or quarantine requirements, in combination with coping internally with this problem, were related to more task changes. In other words, where practices solved or had to solve problems due to absence themselves, they had to reallocate tasks to be able to continue providing care to their patients. The general idea that the opportunities for task changes are greater in larger practices, regarding the number of different disciplines working in the practice, was confirmed. The practice location was not related to task changes. GP practices in city environments as well as those in rural areas were affected by the pandemic. Characteristics of the practice population regarding a more-than-average proportion of elderly and people with a chronic condition were related to task changes; however, the practice composition in terms of other indicators for a vulnerable population was not. 

We also added the evaluation of task changes by GPs to the analysis, without (in the introduction) formulating a specific expectation about the direction of the influence. The evaluation of task changes showed a positive association with task changes; GPs were happy with the task changes in practices where more changes were experienced. At the same time, GPs in these practices felt the need for further training. Coping with the pandemic through improved cooperation between neighboring practices was positively associated with task changes. However, we cannot say anything about the causal direction of these associations.

At the country level, we expected a potential pandemic-specific influence on task changes from the urgency of changes, the role of GPs during the pandemic, and the way GPs are paid. As general country/health system influences, we expected that already realized shifts of task from GPs to other staff—in other words, the point of departure—might play a role, as well as the institutional aspects that indicate how normal it is that former GP tasks are done by nursing staff (in particular, prescription rights as an indicator for this). Finally, we expected that task changes would be easier to implement in strong PC systems. As it turned out, these characteristics at the country level were not associated with task changes. We also assumed that the urgency of task changes may have played a role; this was related to the intensity of the COVID-19 pandemic in each country in terms of, e.g., numbers of infections during the first wave of the pandemic. This idea was close to confirmation in the statistical analysis of the total data set and significant in the sensitivity analysis. We see this as an indication that the urgency of task shifts in a country indeed made a difference in the extent of task changes.

Overall, the extent to which we could statistically explain the variation in task changes at GP and country level was small. This could be related to our measurement of key variables, or to not including key variables that influence task changes. For example, we did not have information about the exact tasks and about organizational procedures before the pandemic. Moreover, there might also have been shortages of GPs and other staff before the pandemic. We also have not included the government’s response to the pandemic. 

This also means that we cannot expect to have found the magic bullet for implementing changes, based on the variables that turned out to be related to task changes. For example, there is a relationship between the availability of support staff and nurses in a practice and task changes [11]. However, although intuitively appealing, we can only say that increasing the number of disciplines would perhaps contribute to task changes, but only to a small extent; other measures not studied in our analysis are perhaps more important. Keeping these nuances in mind, we will discuss the practical implications of our study in the next section.

### 4.2. Implications for Policy and Practice

Practices where more different disciplines are working showed more changes in tasks. This may indicate that these practices can better adapt to new situations [12,22]. The organization of GP practices regarding multi-disciplinarity shows big differences between European countries [9]. This provides opportunities to learn from each other. Based on the experience during the pandemic, developing and evaluating new models of primary care delivery, including the GP and staff task changes, is a challenge for future research. 

An important consideration is whether changes will be sustained after the pandemic. Some of the changes, such as a more outreaching approach to patients by GPs and staff, are generally relevant, given demographic and morbidity changes and the tendency for people to stay longer in their own living environment [23]. It is noticeable that more task changes were found in practices with an older patient population and with more patients with a chronic condition. The experiences during the pandemic with a more outreaching approach could also be used in the normal situation. 

Policy in Spain is to make the changes in tasks more permanent; legal changes are coming. 

In the UK, there was a decrease in home visits (taken over by paramedics who were not trained for this), but also more participation from the community [24]; volunteers came to the practice to help (e.g., in distributing prescriptions) and recently retired clinicians were helping GP practices. In, for example, the Netherlands [25], Belgium [26] and Germany [27], the number of home visits decreased, too. According to these cited authors, the decreases are most probably due to a combination of reduced accessibility because of infection prevention measures and the reluctance of patients to visit the practice.

Changes in the tasks and responsibilities of practice staff require acceptance by patients, and they have to put trust in the competencies of the practice staff. This may require additional investments in the education of practice staff. Quality of care is an important issue to consider when tasks change and, particularly, when tasks are shifted to nurses and support staff. This has been studied extensively in the context of shifting tasks from GPs to nurses, and research has shown that this is possible without compromising the quality of care provided [28,29,30,31,32]. However, it is unclear whether this is also the case with the quicker and forced changes during the pandemic. This requires further research into effectiveness, quality and safety. GPs felt a need for further training in practices with more task changes. Patients seem to have diverging views on online consultations and voiced worries about the quality of care in a Spanish study [33]. With a view to the sustainability of changes in tasks and quality of care, it is important that educational institutions and professional organizations together think of an adequate answer to this need.

Whether changes in tasks are always appropriate and accepted by patients may be questioned. In particular, during times of crisis changes may not be evidence-based or they are introduced without consultation with stakeholders. This may be different from country to country. For example, Spanish PC practices asked urgently for some of the task changes, needing nurse aids (similar to paramedics) to take test samples, or support staff to assume part of the administrative tasks, allowing both nurses and doctors to focus on clinical issues [13]. Moreover, some tasks that were formerly done were no longer or less possible, e.g., nurses doing chronic disease management that was affected by restrictions or even ceased during a lockdown [34].

The COVID-19 pandemic revealed the importance of learning between health systems [3]. Technological and organizational innovations have been implemented at remarkable speed. The changes in staff tasks in GP practices most probably went quicker than normal processes of implementation of task changes, showing how slow our existing processes normally are in comparison. A case in point is the introduction and maintenance of remote consultations [35]. There are still many challenges for health systems to learn from each other [36]. For example, a more proactive approach in outreaching to vulnerable patients differs between countries. This is relevant to PC even apart from the situation during the pandemic, given the changes in demography, the numbers of vulnerable people living in their own homes and changes in morbidity. 

### 4.3. Strengths and Weaknesses

A unique characteristic of the PRICOV-19 study is that it was set up without substantial funding and that countries participated voluntarily. A very strong feature of this study is that no less than 38 countries were willing to participate. This enabled us to use appropriate statistical approaches, in particular multilevel analysis. The number of responding practices differs and is quite low in some countries. However, due to the Bayesian character of multilevel analysis, the countries with a lower number of responses weigh less heavily on the total results through shrinkage of the estimates to the overall mean [19]. 

The response group of participating practices was not representative of the population of practices according to practice location, size and type. There are several reasons for this. First of all, in many countries there is no sampling framework of practices available for researchers. Secondly, the population distribution was asked for from the country partners and in nearly half of the cases they had to make an estimate as actual population data were not available. The estimates may differ from the actual data. Thirdly, respondents may have placed themselves in a category different from the one they would be in for the population data. Fourth, the burden of COVID-19 may have been heavier in smaller practices and those with less human resources. The lack of representativeness cannot be solved by using survey weights, on the one hand because the population distributions were not always based on actual data, and on the other hand because the response groups are often too small to use survey weights. In our view, representativeness is less important in the analysis of associations (as was the focus in this paper) than in descriptions of the situation. For the descriptive aim of this paper, we argue that information on the organization of GP practices during COVID-19, though not fully representative, is better than no information at all. Participation rates and representativeness form a problem in most (international) studies that require the voluntary participation of GPs/practices. In the case of the PRICOV-19 study, the whole study was not separately funded and it was dependent on the voluntary efforts of the country partners and the GPs.

The PRICOV-19 study uses surveys. Therefore, the changes in tasks during the pandemic are based on self-reports. This may include socially desirable answers. Mostly, GPs have answered the surveys. Respondents have also answered questions about the tasks of practice staff. However, staff members may have different perspectives on the changes in tasks. Moreover, the practice staff of general/PHC practices, their actual tasks and competencies, and even the names by which they are known, differ between countries [37,38,39]. In this survey study, this could not be addressed in depth. 

We constructed a scale for task changes that is made up of different changes; this reduces the data but at the same time loses some of the richness. In the analysis, we were not able to differentiate, e.g., between a shift of responsibility or intensification of tasks or changes for practice staff or GPs. 

We used a cross-sectional survey. This restricts the possibilities to make causal inferences. Therefore, we could not make causal inferences; we have only shown associations. The survey was conducted between November 2020 and October 2021. The timing of the survey may have influenced the results. A year and several waves later may have set a different situation. Moreover, in countries with a later start for the data collection, more measures in the practices may have already been taken and more experience with handling the pandemic in PC may have been built.

The data we used for the extent of task changes before the pandemic were already collected ten years before the pandemic. In the meantime, more changes may have taken place. However, there is no recent information about this in an international comparative context.

As an institutional factor that may be related to task shifting, we used whether or not nurses have prescription rights. We have used this variable because of a relationship with task shifting in a previous study [11]. There may be other institutional influences on task changes; however, we have not found comparative data for other variables.

As a higher level, we used countries. However, the organization of PC and the policies regarding COVID-19 may differ within some federal or decentralized countries [40]. There may also be regional differences within countries in the intensity or timing of the pandemic, e.g., in Switzerland [41], or in the availability of primary care workforce [42,43]. We could not take this into account.

## 5. Conclusions

PC has seen many changes during the COVID-19 pandemic. In this paper, we focused on changes in the tasks of practice staff and GPs/trainees. Hereby, much variation was found between practices and countries. Practice staff and GPs have taken up more responsibilities. In addition, practices have become more outreaching to vulnerable patients. This may require further education and training and improved cooperation, internally within practices as well as between practices. Multidisciplinary practices saw more task changes, as did practices with more elderly and more people with chronic conditions. The important issues for the future are which changes can be kept and integrated into practice during normal times.

## Figures and Tables

**Table 1 ijerph-19-15329-t001:** Changes in tasks of practice staff by country (row percentages). Since the COVID-19 pandemic, staff members are more involved in ^a^.

	1. Giving Information by Telephone		2. Explaining Low Literacy		3. Actively Reaching Out to Patients		4. More Involved in Triage		
Country	Agree	Strongly Agree	Agree	Strongly Agree	Agree	Strongly Agree	Agree	Strongly Agree	N between
Austria	45.8	29.0	39.2	9.6	36.4	12.1	46.2	40.2	125–132
Belgium	39.8	45.0	34.4	30.1	26.5	15.9	39.2	50.5	279–291
Bosnia and Herzegovina	32.3	12.9	29.0	12.9	22.9	9.7	32.3	9.7	31
Bulgaria	61.3	26.3	58.0	23.5	45.7	19.8	59.3	28.4	80–81
Croatia	44.2	35.8	47.5	29.7	40.7	20.3	43.0	43.8	118–121
Cyprus	70.0	20.0	30.0	50.0	40.0	40.0	70.0	20.0	10
Czech Rep	43.7	42.7	51.1	24.5	51.0	9.8	44.7	42.7	94–103
Denmark	30.6	52.8	32.4	29.4	47.2	19.4	47.2	38.9	34–36
Estonia	46.2	46.2	42.7	27.0	52.9	12.8	35.6	52.9	89–106
Finland	40.2	4.9	5.9	31.4	4.9	9.8	14.0	5.0	100–102
France	39.3	38.6	25.3	28.3	24.2	17.0	34.7	39.7	367–420
Germany	31.2	66.4	35.5	22.3	40.9	17.3	40.9	46.5	251–254
Greece	65.1	19.8	61.4	19.3	47.7	12.8	55.7	23.8	86–88
Hungary	41.2	48.5	48.6	32.4	35.1	16.2	47.9	41.1	173–194
Iceland	31.1	57.1	37.5	25.0	18.5	3.7	35.7	46.4	24–28
Ireland	59.0	26.0	36.9	11.3	49.1	10.7	48.6	42.9	160–175
Israel	51.5	12.1	18.8	4.7	27.0	9.5	46.0	20.3	64–74
Italy	25.8	69.2	26.8	66.9	31.4	45.5	33.1	56.1	156–159
Kosovo	61.5	26.2	49.3	26.8	49.3	34.3	48.4	25.0	64–67
Latvia	18.1	71.4	20.0	60.0	18.9	66.7	17.7	75.4	105–133
Lithuania	36.5	53.9	36.7	44.9	50.0	22.9	34.6	55.8	48–52
Luxemburg	5.6	25.0	42.9	21.4	26.7	6.7	13.3	76.7	14–15
Malta	42.9	14.3	0.0	14.3	28.6	0.0	42.9	42.9	7
Moldova	58.8	38.2	56.7	35.8	66.7	27.3	49.3	46.3	66–68
Netherlands	38.1	33.8	34.5	21.0	36.9	14.0	34.2	44.9	148–160
North Macedonia	52.5	10.0	60.5	7.9	57.5	10.0	57.9	10.5	38–40
Norway	41.1	37.9	30.0	10.0	34.4	11.2	51.2	23.6	100–127
Poland	41.1	50.0	44.4	34.4	47.9	22.9	44.8	47.9	160–192
Portugal	42.8	44.8	43.3	30.4	52.8	27.2	42.4	43.4	194–201
Romania	41.1	51.1	40.5	48.3	37.4	47.3	37.0	55.4	89–92
Serbia	48.0	12.2	46.7	10.9	40.7	11.6	43.3	19.6	86–98
Slovenia	44.9	32.4	40.6	22.9	40.1	18.6	42.8	37.6	170–176
Spain	45.7	30.6	39.1	21.8	35.9	18.3	47.3	26.0	271–278
Sweden	30.3	67.1	26.4	9.7	48.0	13.3	50.0	31.6	72–76
Switzerland	42.7	46.3	44.9	18.0	24.4	17.1	37.4	51.8	78–83
Turkey	44.7	22.8	41.8	14.8	36.1	17.2	35.2	16.0	122–125
Ukraine	49.8	22.7	46.3	24.1	49.8	14.3	54.5	19.6	216–225
United Kingdom	43.5	30.4	40.9	40.9	60.9	4.3	47.8	39.1	22–23

^a^ Dark grey: highest tertile; light grey: middle tertile; no color: lowest tertile.

**Table 2 ijerph-19-15329-t002:** Changes in the tasks of the respondent (usually a lead GP or GP practice owner) by country (row percentages). Since the COVID-19 pandemic ^a^.

	My Responsibilities Increased		GPs Are More Involved in Reaching Out to Patients		
Country	Agree	Strongly Agree	Agree	Strongly Agree	N between
Austria	- ^b^	- ^b^	43.2	13.6	132
Belgium	37.2	30.1	34.3	15.9	233–440
Bosnia and Herzegovina	34.4	53.1	45.5	15.2	32–33
Bulgaria	41.2	52.9	54.4	17.8	85–90
Croatia	22.4	68.0	40.3	27.1	125–129
Czech Rep	30.5	54.2	49.0	8.3	59–96
Cyprus	33.3	67.7	50.0	20.0	9–10
Denmark	34.5	41.4	30.6	16.7	29–36
Estonia	38.9	36.7	42.6	17.6	90–108
Finland	23.4	18.8	25.3	2.1	64–95
France	31.4	28.0	38.8	13.3	510–528
Germany	35.1	33.5	29.1	7.6	248–251
Greece	25.6	59.0	53.8	17.5	78–80
Hungary	22.5	74.2	31.5	14.0	178–182
Iceland	44.4	18.5	17.9	7.1	27–28
Ireland	35.7	45.0	49.4	13.3	166–171
Israel	40.8	14.3	39.2	10.8	49–74
Italy	47.5	46.8	25.4	67.7	141–201
Kosovo	39.1	48.4	54.1	31.1	64–74
Latvia	14.2	76.4	22.0	63.6	127–132
Lithuania	17.8	77.8	39.6	25.0	45–48
Luxemburg	37.5	25.0	26.7	0.0	15–16
Malta	66.7	22.2	25.0	0.0	8–9
Moldova	35.5	62.9	55.4	32.3	62–65
Netherlands	30.8	31.4	41.0	9.0	144–156
North Macedonia	36.8	0.0	88.1	4.8	41–42
Norway	32.5	27.0	42.5	17.3	126–127
Poland	41.2	49.5	47.6	19.9	182–191
Portugal	27.7	65.6	48.5	22.7	195–198
Romania	19.5	74.4	39.3	49.4	82–89
Serbia	21.6	68.1	35.6	17.3	104–116
Slovenia	50.3	29.5	32.2	6.4	171–173
Spain	29.9	49.3	42.7	20.8	274–278
Sweden	33.3	64.6	27.0	9.5	48–74
Switzerland	0.0	28.6	37.8	7.3	7–82
Turkey	32.5	52.1	32.3	26.6	117–124
Ukraine	45.6	45.2	52.1	23.1	228–234
United Kingdom	40.9	27.3	27.3	9.1	22

^a^ Dark grey: highest tertile; light grey: middle tertile; no color: lowest tertile. ^b^ Question not asked in Austria.

**Table 3 ijerph-19-15329-t003:** Evaluation and satisfaction with changes in GPs’ tasks by country (row percentages) ^a^.

	Happy with the Task Shifting		Do Not Feel Prepared for the Task Shifting		Need Further Training		
Country	Agree	Strongly Agree	Agree	Strongly Agree	Agree	Strongly Agree	N
Austria	- ^b^	-	-	-	-	-	-
Belgium	17.2	3.2	20.5	3.1	17.8	2.1	373–381
Bosnia and Herzegovina	12.9	0.0	28.1	6.3	65.6	6.3	31–32
Bulgaria	9.5	4.7	6.9	0.0	21.4	4.8	84–87
Croatia	19.7	4.1	13.1	4.1	21.0	6.5	122–125
Czech Rep	8.6	6.9	13.8	3.5	24.6	0.0	57–59
Cyprus	11.1	0.0	11.1	0.0	22.2	0.0	9
Denmark	25.0	3.6	24.1	6.9	32.1	7.1	28–29
Estonia	20.5	1.1	18.0	2.3	37.8	2.2	88–90
Finland	27.3	12.7	11.9	1.7	24.1	5.2	55–64
France	13.9	3.2	19.1	5.2	15.4	3.3	487–510
Germany	30.6	5.8	14.2	8.5	15.5	4.1	242–248
Greece	26.7	10.7	21.1	1.3	47.4	7.9	75–78
Hungary	19.7	4.4	18.2	4.0	31.9	6.0	176–183
Iceland	23.1	23.1	7.7	0.0	18.5	0.0	26–27
Ireland	38.3	10.2	9.5	3.0	24.0	2.9	167–171
Israel	25.5	4.3	23.9	4.4	30.4	4.4	46–49
Italy	18.6	11.4	33.6	8.6	47.1	9.4	138–141
Kosovo	44.6	40.0	30.2	19.1	47.7	18.5	63–65
Latvia	19.8	10.3	15.2	4.0	7.2	10.4	125–127
Lithuania	24.4	0.0	9.1	6.8	43.2	11.4	44–45
Luxemburg	20.0	0.0	21.4	0.0	28.6	0.0	14–15
Malta	22.2	11.1	0.0	0.0	22.2	0.0	9
Moldova	25.8	16.1	18.0	3.3	50.0	6.7	60–62
Netherlands	12.8	2.7	14.8	3.4	8.0	2.0	148–156
North Macedonia	25.0	10.0	14.6	9.8	41.5	4.9	40–41
Norway	35.3	5.9	10.6	0.8	19.2	1.7	119–126
Poland	25.4	2.8	16.2	6.7	44.4	5.1	177–182
Portugal	11.9	3.1	18.6	4.2	43.5	10.4	192–195
Romania	27.5	15.0	22.5	8.8	45.0	17.5	80–82
Serbia	31.6	14.5	16.5	6.1	21.9	11.4	114–117
Slovenia	22.4	2.3	9.3	4.1	44.3	8.6	173–174
Spain	18.4	9.4	10.4	2.9	30.5	6.8	277–279
Sweden	19.2	0.0	19.2	8.5	6.4	0.0	47–48
Switzerland	0.0	0.0	16.7	0.0	33.3	0.0	6–7
Turkey	11.4	6.1	19.3	9.7	45.1	15.9	113–117
Ukraine	34.4	10.1	38.0	11.8	53.8	12.6	221–227
United Kingdom	59.1	0.0	22.7	0.0	18.2	0.0	22

^a^ Dark grey: highest tertile; light grey: middle tertile; no color: lowest tertile. ^b^ Question not asked in Austria.

**Table 4 ijerph-19-15329-t004:** Staff absence since the COVID-19 pandemic (average number of staff absent; between brackets: weighted by total number of paid staff), and coping with absenteeism of practice staff (coped with internally, coped with in cooperation with neighboring practices, improved cooperation with neighboring practices) by country (row percentages) ^a^.

	Staff Absence (Weighted)	Coped with Internally		Coped with in Cooperation with Neighboring Practices		Improved Cooperation with Neighboring Practices		
Country	Mean	Agree	Strongly Agree	Agree	Strongly Agree	Agree	Strongly Agree	N
Austria	1.3 (0.19)	33.1	14.5	42.7	18.6	29.1	4.7	124–127
Belgium	1.5 (0.27)	36.1	13.7	34.4	16.9	25.5	9.0	410–443
Bosnia and Herzegovina	4.6 (0.59)	34.5	0.0	37.9	0.0	28.6	0.0	26–29
Bulgaria	0.9 (0.28)	37.5	7.2	41.5	8.5	39.8	8.4	82–83
Croatia	1.2 (0.52)	28.2	8.1	44.8	24.0	39.0	14.6	117–125
Czech Rep	0.9 (0.24)	23.3	3.3	45.4	21.7	24.0	7.3	90–99
Cyprus	2.9 (0.48)	33.3	11.1	40.0	10.0	33.3	11.1	9–113
Denmark	3.4 (0.44)	29.4	2.9	58.8	17.7	20.0	5.7	34–35
Estonia	1.6 (0.16)	45.1	5.9	49.5	5.3	35.0	4.0	95–109
Finland	5.8 (0.10)	29.0	2.0	27.8	6.2	35.5	11.8	85–100
France	1.9 (0.22)	23.6	12.5	22.8	3.7	31.0	11.9	479–500
Germany	2.0 (0.19)	40.7	17.9	42.6	17.8	23.9	3.6	242–247
Greece	4.3 (0.16)	50.0	14.3	35.0	13.8	40.0	8.8	70–84
Hungary	0.7 (0.26)	22.4	5.8	50.0	12.8	40.2	8.9	172–179
Iceland	6.8 (0.23)	45.8	12.5	45.8	16.7	33.3	16.7	23–24
Ireland	2.9 (0.27)	41.8	6.1	23.3	7.6	39.2	7.2	159–166
Israel	4.9 (0.30)	30.0	2.9	45.7	0.0	20.9	0.0	67–70
Italy	0.4 (0.10)	16.2	4.6	6.1	1.0	4.6	0.5	198–200
Kosovo	36.9 (0.45)	40.3	50.0	31.4	47.1	40.0	45.7	46–72
Latvia	1.0 (0.26)	22.9	53.4	30.8	31.6	24.6	15.6	117–122
Lithuania	16.9 (0.13)	35.3	31.4	15.6	11.1	18.8	4.2	45–51
Luxemburg	2.21 (0.31)	26.7	6.7	28.6	7.1	64.3	14.3	14–53
Malta	3.5 (0.29)	50.0	0.0	42.9	0.0	16.7	0.0	6–8
Moldova	21.8 (0.25)	47.6	30.2	40.3	17.7	38.7	22.6	62–64
Netherlands	4.1 (0.34)	43.1	24.2	36.7	30.0	40.1	17.8	150–153
North Macedonia	2.0 (0.51)	64.7	2.9	69.0	3.4	48.5	6.1	29–34
Norway	2.6 (0.25)	32.8	5.0	24.1	3.5	11.9	1.7	34–119
Poland	4.2 (0.34)	39.9	9.0	20.4	3.0	22.1	2.3	167–178
Portugal	5.9 (0.24)	25.8	10.8	14.9	1.7	33.3	6.1	180–186
Romania	0.7 (0.24)	44.0	14.7	48.1	9.1	44.2	5.2	75–81
Serbia	15.9 (0.36)	34.2	15.4	22.2	9.3	31.2	8.3	84–117
Slovenia	0.8 (0.24)	34.8	6.8	48.5	31.5	44.1	13.0	161–166
Spain	10.2 (0.24)	9.1	1.1	7.92	0.8	10.8	1.9	260–266
Sweden	16.6 (0.36)	40.3	8.3	25.0	1.4	52.1	15.5	70–72
Switzerland	2.1 (0.19)	43.6	11.5	33.8	5.4	11.4	3.8	74–80
Turkey	4.0 (0.39)	42.6	16.5	13.2	4.4	21.2	2.7	104–115
Ukraine	25.8 (0.44)	32.7	8.2	36.0	7.4	35.8	6.7	193–208
United Kingdom	9.7 (0.36)	28.6	9.5	28.6	4.8	61.9	28.6	21

^a^ Dark grey: highest tertile; light grey: middle tertile; no color: lowest tertile.

**Table 5 ijerph-19-15329-t005:** Multilevel analysis of task changes in GPs, trainees and practice staff.

	Model 1: Empty Model Coefficient (SE)	Model 2: Practice Variables Coefficient (SE)	Model 3: Interaction Terms Coefficient (SE)	Model 4: Country Variables ^c^ Coefficient (SE)
*Fixed part*				
Constant	2.822 (0.053)	2.472 (0.068)	2.485 (0.068)	
Staff absence		−0.001 (0.001)	−0.004 (0.001) *	
coped with internally ^b^		−0.008 (0.007)	−0.014 (0.007)	
coped with neighboring practices		−0.014 (0.008)	−0.013 (0.008)	
improved cooperation with neighboring practices ^a,b^		0.070 (0.009) **	0.070 (0.009) **	
GPs happy with the task shifting		0.066 (0.008) **	0.065 (0.008) **	
GPs do not feel prepared		0.014 (0.009)	0.013 (0.009)	
Need further training		0.021 (0.009) *	0.021 (0.009) *	
Practice size		1.02 × 10^−6^ (9.82 × 10^−7^)	1.03 × 10^−6^ (9.87 × 10^−7^)	
Number of GPs and trainees		−0.000 (0.000)	−0.000 (0.000)	
Total number of paid staff		0.000 (0.002)	0.001 (0.002)	
Number of disciplines ^a,b^		0.009 (0.004) *	0.009 (0.004) *	
GPs paid (mixed) fee-for-service		−0.027 (0.025)	−0.028 (0.025)	
Practice location (ref. big city)				
-suburbs		−0.028 (0.028)	−0.028 (0.028)	
-(small) towns		−0.018 (0.023)	−0.017 (0.022)	
-mixed urban–rural		0.005 (0.022)	0.005 (0.022)	
-rural		−0.020 (0.024)	−0.020 (0.024)	
Practice population elderly/chronic conditions		0.017 (0.007) *	0.016 (0.007) *	
Practice population other vulnerable populations		0.003 (0.003)	0.003 (0.003)	
Interaction staff absence * coping internally			0.001 (0.000) **	
Interaction staff absence * coping neighboring practices			0.005 (0.008)	
Interaction staff absence * coping improved cooperation			−0.000 (0.000)	
COVID-19 cases per million population during 1st wave				0.008 (0.005) (*p* = 0.107)
Idem COVID-19 mortality				−0.004 (0.005)
COVID-19 cases per million population 3 months before survey				−0.001 (0.005)
Idem COVID-19 mortality				0.006 (0.005)
Role of GPs during pandemic				0.024 (0.036)
Strength of PC				−0.061 (0.398)
Nurse prescribing rights (yes)				−0.146 (0.119)
Degree of task shifting in 2012				−0.024 (0.056)
*Random part*				
Country variance	0.10 (0.024)	0.10 (0.025)	0.10 (0.025)	
Practice variance	0.29 (0.006)	0.28 (0.006)	0.28 (0.006)	
ICC (%)	25.7	27.0	26.9	

* *p* < 0.05. ** *p* < 0.01. ^a^ The missing-value indicator for this variable is significant in Model 2. ^b^ The missing-value indicator for this variable is significant in Model 3. ^c^ Country variables have been added one by one; coefficients of practice-level variables are not reported—they differ only marginally from those in Model 3.

## Data Availability

The anonymized data is held at Ghent University and is available to participating partners for further analysis upon signing an appropriate usage agreement.

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
