# Peer review of "Has the COVID-19 Pandemic Led to Changes in the Tasks of the Primary Care Workforce? An International Survey among General Practices in 38 Countries (PRICOV-19)"

_ijerph, 2022, doi:10.3390/ijerph192215329_

Round 1

Reviewer 1 Report

The paper deal with an interesting topic because many changes had happened during the COVID-19 pandemic.

The quality of presentation is good and the paper is scientifically sound. 

Some minor issue could be improved.

Please revise the text for minor spell check.

In the discussion the A.A. discuss on task change, does the A.A. think that this modification could also be related to modified patient's habit?

The A.A. could discuss with the finding of 

Famà F, Lo Giudice R, Di Vita G, Tribst JPM, Lo Giudice G, Sindoni A. COVID-19 and the impact on the cranio-oro-facial trauma care in Italy: An epidemiological retrospective cohort study. Int J Environ Res Public Health 2021;18(13). 

Reviewer 2 Report

Many thanks for giving me the opportunity to discover this very interesting international study. This study was undertaken early during the pandemic and collected highly relevant data from 38 countries.

However this paper may present several important limitations.

Major comments

The definition of the outcome of interest « changes in the tasks » remains quite inintelligible for me. First, I didn't understand exactly how the 6 items were used to lead to the scale (l. 275). Most importantly, it seems that the authors considered the differents types of changes (GPs or stafff, changes related to giving information or to increased responsability...) as a continuous (or at least ordinal ?) variable. It seems to me that this is a very strong (and quite uncertain) hypothesis. Authors only briefly discuss it l. 570-572. Finally, it is quite difficult to interpret « changes » without taking into account the baseline level (not only as an independent variable among others).

Distribution of the scales (4-item staff ; 2-item GPs ; 6-item global scale) between countries may be provided to make it easier for the reader to follow the paper. As well as some bivariate analyses before regression.

In the other side, the regression process to model « changes in the tasks » seems potentially problematic since :

-          It is not clear which variables should be considered as dependent or independent. For example, it is questionable to consider « GPs evaluation of their role changes » or « need for training » as independent variables… and changes itself as a dependent variable !

-          Some of the (numerous !) independent variables are expected to be collinear. For example, strength of PC / task shifting around 2011 / nurses prescription rights / number of disciplines, etc. Same remark for variables at the practice level. No analysis of colinearity is provided.

-          Some variables may lack variability within some countries, with implications for the modeling process. For example, what about the distribution of number of disciplines among practices in countries where 1) single-handed practices are the rule or 2) multidisciplinary teams are the rule ?

Most of the variables considered are of great interest for the research question but I wonder whether a regression model is appropriate for these data / for the objective of the authors. As a suggestion, a multidimensional data analysis may be more appropriate 1) to better apprehend the richness of the data – related to tasks changes - (rather reducing it, see l. 570-572) as well as 2) to lead to results easier to interpret : how can the reader interpret results from a scale that he/she is not sure to know what exactly is measures ?

The discussion strongly lacks bibliographic references

Minor comments

As it may constitute an important limitation for the data (from nov 2020 to oct 2021 !), period of survey in countries should be at least provided to the reader. Were sensitivity analyses undertaken to see whether survey / participation period was associated with results (at practice / country levels) ?

What is professionalisation ? (additional table 2)

In which proportion did GPs themselves participate versus practice staff ? Differences between countries ? with potential influence on the answers/ results ?

It seems that practice staff were more often involved in giving information / recommandation to patients contacting the pratice and in the triage process (=> « reactive » tasks ?), rather than being more often involved beside vulnerable populations (low literacy / migrants etc) or in actively reaching out to patients (=> proactive » tasks ?). How do the authors interpret it ? This point would be interesting to develop and to confront to existing literature.

The abstract clarity may be improved. It is difficult to understand without reading the whole paper.

The introduction is rich and provides important background in terms of concepts used (for example tasks changes vs taks shifting) and in terms of data and underlying hypotheses. However introduction, as well as the text in general, is very long and often has redundancies.

Some variables provided at a country level may be not so relevant at this scale. The authors partly address this point (l. 588-589) but other variables / limitations may be cited such as territorial differences within countries in terms of: intensity of the pandemic ; shortage of primary care workforce, etc.

The tables are very long and sometimes legends are lacking (for example, table 4 : absenteism : what period was considered? Most importantly : scores used in additional table 2 are not exaplined : role of GPs, professionalisation, Strength of PC…). In addition, shading is not appearing in the tables (perhaps a technical problem ?)

In conclusion, although the data presented in this paper has a high potential to inform tasks changes in GPs practices early during the pandemic including differences between countries, I would suggest important revisions in both substance and form before publication.

Reviewer 3 Report

This article has the same problem as previous ones based on the PRICOV-19 study: the possibility that the sample is representative of the universe of European general practitioners. This issue is not well addressed in this document, nor in previous publications.

Round 2

Reviewer 3 Report

It is unusual to address a research weakness with an “article to be published”. I consider that the information that supports the representativeness of the sample must be available prior to the publication of this work. If it does not appear in the same issue of the journal, it should at least be available as a preprint.

Author Response

Reviewer's comment:

It is unusual to address a research weakness with an “article to be published”. I consider that the information that supports the representativeness of the sample must be available prior to the publication of this work. If it does not appear in the same issue of the journal, it should at least be available as a preprint.

Authors’ reaction:

We thank the reviewer for this quick reaction. We agree that it is unusual to refer to an unpublished paper. Our aim is that the paper on recruitment and participation will appear in the same special collection of papers on the PRICOV-19 study with this paper. However, we cannot guarantee this. We have therefor changed the manuscript and added additional information.

In the methods sub-section on sampling and recruitment we have replaced lines 197-199 by:

‘The country partners provided additional information on the recruitment strategies and sampling procedures. Moreover, they gave (estimates) of the population distribution according to urbanisation of the practice location (big city, suburbs and small towns, semi-rural and rural), the size of the practice (number of patients in five categories) and the practice type (solo, duo, group) in order to be able to assess representativeness of the participating GPs.’

In the methods sub-section on statistical analysis we have added the following in lines 279-283: 

‘To examine how well the distribution of total the population of GP practices according to practice size, location, and type for each country, as reported by country coordinators, reflected the distribution of these characteristics in the participating GP practices in the PRICOV-19 study, we used the standard approach to conduct an one-sample chi-square test [19].’ 

[Reference 19: Parke CS. Module 1: Checking the Representativeness of a Sample. In: Essential First Steps to Data Analysis: Scenario-Based Examples Using SPSS. 2013. SAGE Publications Inc.]

We have replaced the first paragraph of the results section by the following:

‘The recruitment strategies used by the country partners include an invitation published on a website or social media (2 countries), direct contact to all GP practices (12 countries), contact with a sample of GP practices (10 countries), other strategies (3 countries), and multiple recruitment strategies (11 countries). Random sampling was the preferred approach of the study protocol, but in only 6 countries this was realised; in 21 countries a convenience sample was used, and in 11 countries a mix of sampling approaches was used.

Participation rates and representativeness are discussed in the protocol paper (in Table 1 and additional file 3 respectively) [6], and in [23]. The participation rate was 22% (IQR 10-28), varying from below 10% in amongst others Denmark, Sweden and Latvia, to 90% or higher in Serbia, North Macedonia, Greece, and Bulgaria. Overall, the participating practices did not mirror the population of practices in the countries in terms of the distribution by urbanisation of the practice location, the size of the practice and practice type. Practices in small towns and suburbs were under-represented and those in (semi-)rural areas were over-represented, as were large practices (over 10,000 patients) and group practices.’

After these changes we think that the way we worded the limitation in the subsection on strengths and limitations in the discussion from line 636 onwards is justified.

We hope these adaptations sufficiently respond to the reviewer’s comment.